# Hard Prompts Made Easy: Gradient-Based Discrete Optimization for Prompt Tuning and Discovery

**Yuxin Wen**\*, **Neel Jain**\*, **John Kirchenbauer**
University of Maryland
{ywen, njain17, jkirchen}@umd.edu

**Micah Goldblum**
New York University
goldblum@nyu.edu

**Jonas Geiping, Tom Goldstein**
University of Maryland
{jgeiping, tomg}@umd.edu

## Abstract

The strength of modern generative models lies in their ability to be controlled through prompts. Hard prompts comprise interpretable words and tokens, and are typically hand-crafted by humans. Soft prompts, on the other hand, consist of continuous feature vectors. These can be discovered using powerful optimization methods, but they cannot be easily edited, re-used across models, or plugged into a text-based interface.

We describe an easy-to-use approach to automatically optimize hard text prompts through efficient gradient-based optimization. Our approach can be readily applied to text-to-image and text-only applications alike. This method allows API users to easily generate, discover, and mix and match image concepts without prior knowledge of how to prompt the model. Furthermore, using our method, we can bypass token-level content filters imposed by `Midjourney` by optimizing through the open-sourced text encoder. Code is available at `https://github.com/YuxinWenRick/hard-prompts-made-easy`.

## 1 Introduction

Prompt engineering is the art of creating instructions to guide generative models. It is the key to unlocking the power of large models for both image generation and language tasks. As it stands today, prompt engineering methods can be coarsely divided into two camps. First, there are *hard* prompting methods, which use hand-crafted sequences of interpretable tokens to elicit model behaviors. Hard prompt discovery is a specialized alchemy, with many good prompts being discovered by trial and error, or sheer intuition. Then there are *soft* prompts, which consist of continuous-valued language embeddings that do not correspond to any human-readable tokens. Soft prompt discovery is a mathematical science; gradient-based optimizers and large curated datasets are used to generate highly performant prompts for specialized tasks.

Despite the difficulty of engineering hard prompts, they have their advantages. Hard prompts and the tricks they exploit can be mixed, matched, and mutated to perform a range of different tasks, while soft prompts are highly specialized. Hard prompts can be edited by hand to change their behavior. Hard prompts are portable; they can be discovered using one model and then deployed on another. This portability is impossible with soft prompts due to differences in embedding dimension and representation space between models. Finally, hard prompts can be used when only API access to a model is available and it is not possible to control the embeddings of inputs.

---

\*Equal contribution.

37th Conference on Neural Information Processing Systems (NeurIPS 2023).

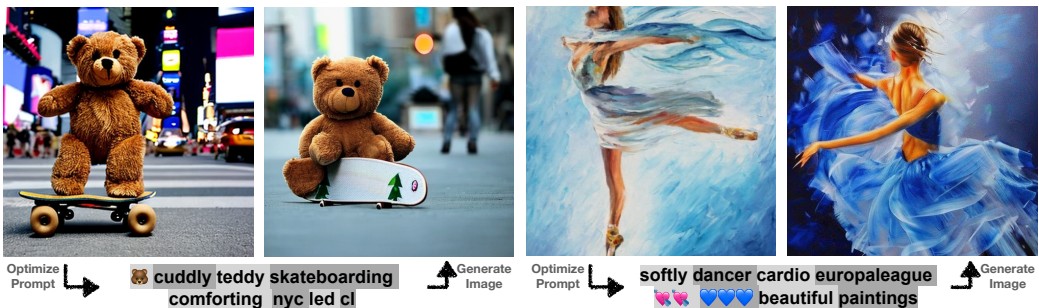

Figure 1: Two examples of hard prompt discovery through optimization. Given an image (left), a discrete text prompt is discovered using CLIP and used to prompt Stable Diffusion, generating new images (right). Two shades of gray are used to show the token boundaries in the recovered prompt.

This work explores the use of efficient gradient methods to optimize and learn discrete text, with an emphasis on applications to prompt engineering. In doing so, we unlock the ability to learn hard prompts via optimization. Learned hard prompts combine the ease and automation of soft prompts with the portability, flexibility, and simplicity of hard prompts. Our primary contributions are summarized as follows:

- We propose a simple scheme for learning hard prompts using continuous optimization. The scheme builds on existing gradient reprojection schemes for optimizing text, and adapts lessons learned from the large-scale discrete optimization literature for quantized networks.

- We show that this optimization method can be used to learn hard prompts for image generation, giving us a general tool to create prompts that elicit specific image styles, objects, and appearances. The learned prompts perform competitively with highly specialized prompt generation tools, despite using far fewer tokens and containing no hand-crafted components.

- We demonstrate that optimizing solely over the text encoder can effectively bypass content filters that use word or token-level red lists. Specifically, our approach successfully circumvents content filters imposed by `Midjourney`, highlighting the need to implement content filters in feature space. Additionally, this suggests that diffusion models may possess a "secret" language, emphasizing the importance of further exploration in this area.

In addition to capturing the quantifiable benefits of learned prompts, the proposed schemes can be used to facilitate *prompt exploration and discovery*, as optimization often recovers words and tokens that are simultaneously highly interpretable and also highly non-obvious.

## 2 Related Works

**Prompting in Language Models.** Brown et al. [2020] was one of the first to demonstrate the power of prompting for task adaption of pre-trained language models. This "instruction tuning" paradigm has since become a standard way to increase the ability of large models to follow complex, task-specific instructions [Sanh et al., 2022, Chung et al., 2022]. However, automatically finding suitable sets of text prompts, i.e. hard prompts, for these purposes remains an open challenge. Lester et al. [2021b] simplified the "prefix tuning" technique presented in Li and Liang [2021] to establish the procedure referred to as standard *soft* "prompt-tuning" where they optimize sequences of continuous-valued embeddings prepended to the real embeddings of the input tokens. However, subsequent work by Khashabi et al. [2022] showed that the sequences of embeddings produced by this technique could map to token sequences with limited semantic scrutability. To address these limitations, in this work we construct a method for hybridizing the continuous soft-prompt optimization with hard vocabulary constraints, resulting in task-specific, interpretable tokens.

**Hard Prompt Optimization.** *AutoPrompt* [Shin et al., 2020] was one of the first discrete prompt optimization frameworks for transformer language models and subsequent approaches have included a gradient-free phrase editing method [Prasad et al., 2022], an embedding optimization approach based on Langevin dynamics [Shi et al., 2022], a reinforcement learning approach [Deng et al.,

2022] and an optimization procedure of learning the target distribution with a continuous matrix of coefficients [Guo et al., 2021].

*AutoPrompt*, which utilizes *HotFlip* proposed by Ebrahimi et al. [2018], greedily chooses the optimal token for each location in the prompt utilizing the gradient to find a selection of good candidates. *FluentPrompt* differs from *AutoPrompt* by utilizing Langevin dynamics [Kumar et al., 2022] to optimize the prompt embeddings, as well as adding a fluency penalty. We consider these two gradient-based methods as baselines, which can be found in supplementary material. However, *AutoPrompt* can become expensive very quickly. For each gradient step, the method requires an evaluation of each candidate at each location in the prompt, adding numerous additional forward passes. To avoid the additional forward passes, we originally considered $AutoPrompt_{k=1}$ with and without an added fluency constraint but found that $AutoPrompt_{SGD}$ with a fluency constraint outperformed its counterparts, and thus we use SGD version of *AutoPrompt* as our other baseline similar to Shi et al. [2022].

For these methods discussed above, at the end of every update step, the optimized prompt embeddings are projected onto their nearest neighbor embeddings to ensure that optimization is performed on the discrete set of natural language tokens. However, if the nearest neighbors are far away from the embeddings and the learning rate is not tuned properly, the embeddings may become stagnant, which can require extensive hyperparameter tuning as demonstrated in Figure 5(a). The cost of such a constraint is a loss of flexibility in the solutions the optimization can find. On the other hand, while soft prompts are not as limited in this way, just clamping a well-trained soft prompt to the nearest discrete prompt strongly degrades performance as observed in Khashabi et al. [2022].

**Prompt Discovery from Images.** The process of extracting rich information from images and conveying it through natural language texts is known as *image captioning*. Zhang et al. [2021], Hu et al. [2022], and Li et al. [2022] achieve this goal by training large captioning models on image-text pairs. However, these captions are often generic and may not accurately reflect new or unseen objects. In Gal et al. [2022], the authors propose a method that utilizes a soft prompt to optimize a text-guided diffusion model, allowing for the generation of similar visual concepts to those in the original image. In this case, though the final soft prompt is effective, optimization through a diffusion model is very expensive, and the prompts are neither interpretable nor portable.

**Discrete Optimization.** Discrete optimizers have long been used to train neural networks with quantized (e.g. binary) weights. In that context, the approach of re-projecting between gradient steps is known as *stochastic rounding*. However, it is known that this approach lacks the convergence guarantees of continuous optimization [Li et al., 2017]. Over the last decade, stochastic rounding has been replaced by newer optimizers that maintain a continuous, rather than discrete, representation of the weights [Courbariaux et al., 2015]. These optimizers consistently result in higher accuracy [Rastegari et al., 2016, Courbariaux et al., 2016] and avoid local minima [Li et al., 2017].

We take inspiration from these lessons learned in the binary networks community and adapt them to refine and simplify discrete optimizers for language.

## 3 Methodology

**Learning Hard Prompts.** We now present our effective and easy-to-use technique for discrete prompt optimization. The process requires the following inputs: a frozen model, $\theta$, a sequence of learnable embeddings, $\mathbf{P} = [\mathbf{e_i}, ...\mathbf{e_M}], \mathbf{e_i} \in \mathbb{R}^d$, where $M$ is the number of "tokens" worth of vectors to optimize, and $d$ is the dimension of the embeddings. Additionally, we employ an objective function $\mathcal{L}$. The discreteness of the token space is realized using a projection function, $\text{Proj}_{\mathbf{E}}$, that takes the individual embedding vectors $\mathbf{e_i}$ in the prompt and projects them to their nearest neighbor in the embedding matrix $E^{|V| \times d}$ where $|V|$ is the vocabulary size of the model, and we denote the result of this operation as $\mathbf{P}' = \text{Proj}_{\mathbf{E}}(\mathbf{P}) := [\text{Proj}_{\mathbf{E}}(\mathbf{e_i}), ...\text{Proj}_{\mathbf{E}}(\mathbf{e_M})]$. Additionally, we define a broadcast function, $\mathcal{B} : \mathbb{R}^{(M \times d)} \to \mathbb{R}^{(M \times d \times b)}$ that repeats the current prompt embeddings ($\mathbf{P}$) in the batch dimension $b$ times.

Formally, to learn a hard prompt, we minimize the following risk by measuring the performance of $\mathbf{P}$ on the task data: $R(\mathbf{P}') = \mathbb{E}_D(\mathcal{L}(\theta(\mathcal{B}(\mathbf{P}, \mathbf{X})), \mathbf{Y}))$.

**Our Method.** We propose a simple but efficient gradient-based discrete optimization algorithm that combines the advantages of the baseline discrete optimization methods and soft prompt optimization.

---

**Algorithm 1** Hard **P**rompts made **E**a**Z**y: PEZ Algorithm

---

**Input:** Model $\theta$, vocabulary embedding $\mathbf{E}^{|V|}$, projection function Proj, broadcast function $\mathcal{B}$, optimization steps $T$, learning rate $\gamma$, Dataset $D$

Sampled from real embeddings:

$\mathbf{P} = [\mathbf{e_i}, ...\mathbf{e_M}] \sim \mathbf{E}^{|V|}$

**for** $1, ..., T$ **do**

    Retrieve current mini-batch $(X, Y) \subseteq D$.

    Forward Projection:

    $\mathbf{P}' = \text{Proj}_{\mathbf{E}}(\mathbf{P})$

    Calculate the gradient w.r.t. the *projected* embedding:

    $g = \nabla_{\mathbf{P}'} \mathcal{L}_{\text{task}}(\mathcal{B}(\mathbf{P}', X_i), Y_i, \theta)$

    Apply the gradient on the *continuous* embedding:

    $\mathbf{P} = \mathbf{P} - \gamma g$

**end for**

Final Projection:

$\mathbf{P} = \text{Proj}_{\mathbf{E}}[\mathbf{P}]$

**return P**

---

The steps of our scheme, which we call PEZ, are concretely defined in Algorithm 1. The method maintains continuous iterates, which in our applications corresponds to a soft prompt. During each forward pass, we first project the current embeddings $\mathbf{P}$ onto the nearest neighbor $\mathbf{P}'$ before calculating the gradient. Then, using the gradient of the discrete vectors, $\mathbf{P}'$, we update the continuous/soft iterate, $\mathbf{P}$.

## 4 Prompt Inversion with CLIP

We showcase the strength of PEZ by learning hard prompts on multimodal vision-language models. With these models, like CLIP [Radford et al., 2021], we can use PEZ to discover captions which describe one or more target images. In turn, these discovered captions can be deployed as prompts for image generation applications. Since most text-guided diffusion models utilize pre-trained text encoders, such as the CLIP text encoder, and freeze them during training, we can discover prompts using these pre-trained text encoders that are directly relevant for downstream diffusion models. For instance, we can optimize a caption which describes an image and use this caption as a prompt for a diffusion model to generate other images with the same content.

Since the CLIP model has its own image encoder, we can leverage it as a loss function to drive our PEZ method. This way we are optimizing prompts only for their cosine similarity to the CLIP image encoder, and avoiding gradient calculations on the full diffusion model altogether.

Formally, given a text encoder function $f$ and an image encoder function $g$, we optimize the hard prompt embedding $\mathbf{P}$ corresponding to a target image $x$ by minimizing the following objective: $\mathcal{L}(\mathbf{P}, x) = 1 - \mathcal{S}(f(\mathbf{P}), g(x))$, where $\mathcal{S}$ is the cosine similarity between two vectors.

### 4.1 Experimental Setting

We conduct experiments on four datasets with diverse distributions: LAION [Schuhmann et al., 2022], MS COCO [Lin et al., 2014], Celeb-A [Liu et al., 2015], and Lexica.art [Santana, 2022]. LAION comprises over 5 billion in diverse images scraped from the internet, including photos and paintings. MS COCO mainly contains real-life photographs with multiple common objects, whereas Celeb-A consists of celebrity portraits. Lexica.art is a set of AI-generated paintings with their prompts.

We measure the quality of the prompt via image similarity between the original (target) image, and an image generated using the learned hard prompt. To do so, we use a larger reference CLIP model, OpenCLIP-ViT/G, that was not used during optimization and serves as a neutral metric for the semantic similarity between the images.

We choose Stable Diffusion-v2 [Rombach et al., 2022] as our generative model, and the open-source CLIP model, OpenCLIP-ViT/H [Cherti et al., 2022] for crafting the prompt, as both share the same text encoder. During the prompt optimization process, we use a generic learning rate of $0.1$ and run 3000 optimization steps using the AdamW optimizer [Loshchilov and Hutter, 2017]. For Stable Diffusion-v2, we set the guidance scale to 9 and the number of inference steps to 25. For each dataset,

Target Image          Generated Images with Learned Hard Prompts
                      Stable Diffusion                    `Midjourney`

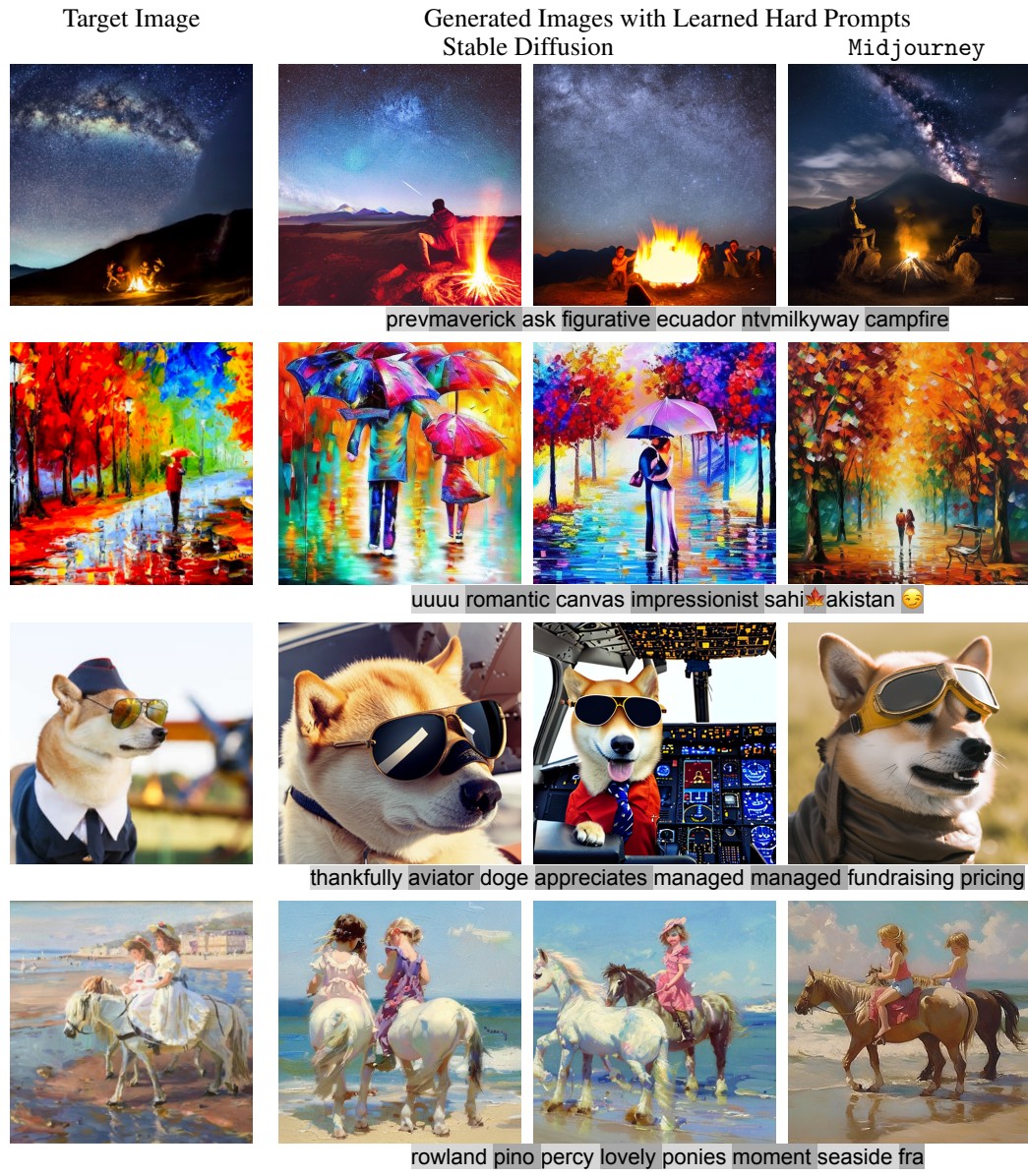

Figure 2: **Generations using learned hard prompts** on four different target images. For a given target image (left), a discrete text prompt is discovered using CLIP and used to prompt Stable Diffusion and `Midjourney`, generating new images (right). Two shades of gray are used to show the token boundaries in the recovered prompt.

we randomly sample 100 data points and average CLIP scores over 5 runs with different random seeds. All experiments are conducted on a single NVIDIA RTX A4000.

A natural baseline for hard prompt discovery with CLIP is the *CLIP Interrogator*[1]. To generate a descriptive hard prompt, this tool first uses a pre-trained captioning model, BLIP [Li et al., 2022] to create a caption of the target image. Then, top-$k$ keywords from a pre-collected bank of keywords are appended to the caption based on CLIP scores between the keywords and the target image. These keywords were collected from various sources, including 5,265 artist names like "Van Gogh" and 100,970 phrases from prompt engineering, resulting in a diverse set. We find this keyword bank to contain most of the phrases from the Lexica.art dataset. *CLIP Interrogator* then greedily samples keywords until the prompt reaches CLIP's token length limit of 77.

---

[1] `https://github.com/pharmapsychotic/clip-interrogator`

Table 1: Quantitative evaluation of learned hard prompts. We report the CLIP score between the original images and the images generated by the hard prompts. A high score indicates that generated and target images contain similar semantic content.

| Method | #Tokens | Requirement | LAION | MS COCO | Celeb-A | Lexica.art |
|---|---|---|---|---|---|---|
| AutoPrompt$_{SGD}$ | 8 | CLIP | 0.689 | 0.669 | 0.595 | 0.702 |
| FluentPrompt | 8 | CLIP | 0.688 | 0.671 | 0.583 | 0.702 |
| PEZ (Ours) | 8 | CLIP | 0.697 | 0.677 | 0.602 | 0.711 |
| CLIP Inter. | $\sim 77$ | C. + Ba. + BL. | 0.707 | 0.690 | 0.558 | 0.762 |
| PEZ + Bank | 8 | CLIP + Bank | 0.702 | 0.689 | 0.629 | 0.740 |
| PEZ + 5 Seeds | 8 | C. + 5 Seeds | 0.705 | 0.692 | 0.614 | 0.722 |
| C. I. w/o BLIP | $\sim 77$ | CLIP + Bank | 0.677 | 0.674 | 0.572 | 0.737 |
| CLIP Inter. | 8 | C. + Ba. + BL. | 0.539 | 0.575 | 0.360 | 0.532 |
| CLIP Inter. | 16 | C. + Ba. + BL. | 0.650 | 0.650 | 0.491 | 0.671 |
| CLIP Inter. | 32 | C. + Ba. + BL. | 0.694 | 0.663 | 0.540 | 0.730 |
| Soft Prompt | 8 | CLIP | 0.408 | 0.420 | 0.451 | 0.554 |

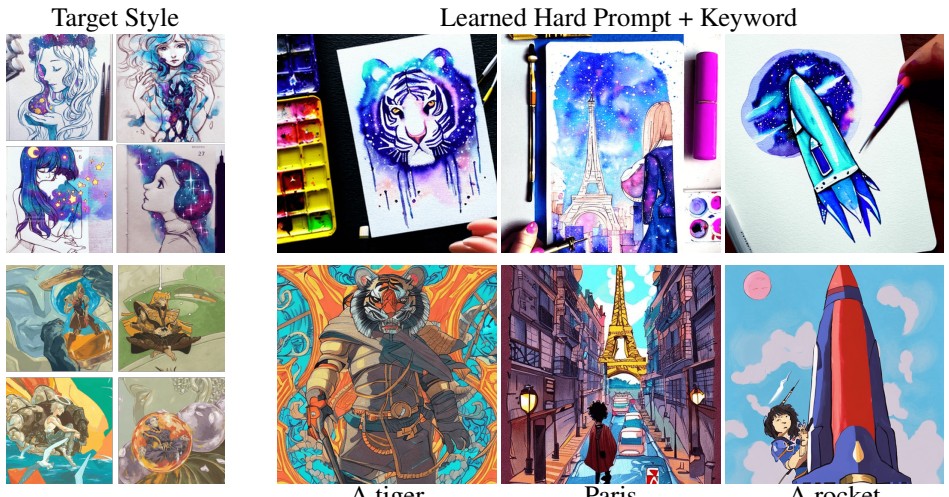

Figure 3: **Learned hard prompts for style transfer**. Given several samples with the same style, we can extract the style with a hard prompt and transfer it to other objects or scenes. Detailed templates and hard prompts can be found in Appendix A.1. Sample images credits: Qinni and facundo-lopez.

## 4.2 Results

We show example hard prompts learned using our method and corresponding generations in Figure 2. The generated images clearly show that the prompts effectively capture the semantic features of the target images. Further, the generations are highly similar to the original images as measured by CLIP score and under visual inspection. Additionally, the hard prompts do not overfit to the original target image and produce a diverse set of generated images given different random seeds.

Prompts are human-readable, containing a mix of real words and gibberish (non-word token sequences). However, the valid words that are included in the prompts provide a significant amount of information about the image. For example, in the first row, we can see the words "milkyway" and "campfire," which are the two main elements in the target image. Interestingly, the optimized prompts may also include emojis, like 🍁 present in the second row. 🍁 represents the trees on the side and also the color theme of the image. The optimization process seems to choose these emojis to include useful information while keeping the prompt concise.

Meanwhile, PEZ is able to find some "secret languages". For instance, in Appendix Figure 10, the prompt "translucent abyss assaulted surfing featured regrann nbappinterest" produces an image of a surfer in a wave tunnel. We find that some tokens like "nbappinterest" and "assaulted" are not

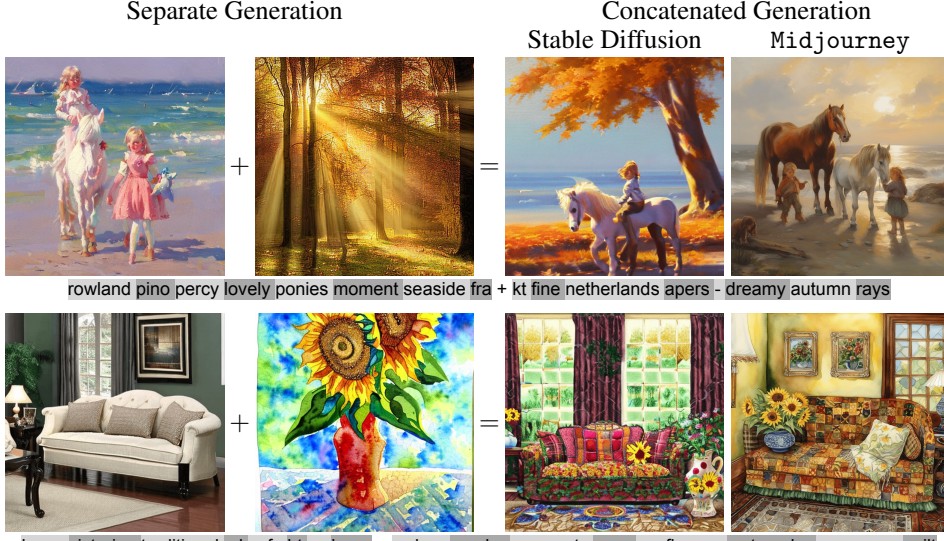

Separate Generation · Concatenated Generation · Stable Diffusion · Midjourney

rowland pino percy lovely ponies moment seaside fra + kt fine netherlands apers - dreamy autumn rays

bway victorian traditional yd sofa ht vn hung + wahoo gumbo payments vase sunflowers watercolor expresses quilt

Figure 4: **Concatenated learned hard prompts**. We show the hard prompts learned on two unrelated images can be concatenated to fuse the semantic concepts in them.

necessary for effective image generation. However, "regrann", although seemingly unrelated to the depicted image and not being a recognized word, is indispensable. It plays a crucial role in generating coherent images, ensuring, for example, that there isn't a second person or that the surfboard remains intact. It should be noted that the term "secret languages" here differs from the definition provided by Daras and Dimakis [2022]. Daras and Dimakis [2022] describe "secret languages" that are entirely unintelligible to humans. However, "secret languages" we talk about here are the words/tokens that contribute to the image in ways that are non-obvious at first glance.

Further, we present quantitative evaluations in Table 1. PEZ performs consistently well across all four datasets and outperforms other gradient-based optimization baselines (full table can be found in the supp. material Table 2). Notably, we can achieve similar performance to *CLIP Interrogator*, which has the highest CLIP score on LAION, MS COCO, Lexica.art, but not Celeb-A (The keyword bank in *CLIP Interrogator* does not include many words related to real human faces). However, *CLIP Interrogator* uses a large curated prompt dataset, the image captioning model BLIP, and a large number of tokens (as many as 77), while our proposed method only uses the CLIP model for prompt discovery and 8 tokens in total demonstrating its simultaneous simplicity and strength.

We ablate each of these differences. To do so, we include the keyword bank in our optimization method and only allow projections onto tokens from the keyword bank. Overall, we find that when adding this constraint to our model, and disabling BLIP to compare both methods on equal footing, we recover most of the quantitative difference between the methods on LAION and Lexica.art. Additionally, reducing the token length for the *CLIP Interrogator*, leads to a sharp drop in performance, again, particularly when normalizing by comparing both approaches at equal token lengths of 8.

In a realistic scenario, our method can be applied iteratively to find better prompts. For each image, we simulate this by optimizing prompts with 5 different initializations and selecting the one with the lowest loss. According to Table 1, this simple technique improves single trial performance and narrows the gap between PEZ and *CLIP Interrogator*.

We note that even though Stable Diffusion and CLIP share the same text encoder, *soft prompts do not transfer well* compared to all hard prompt methods in our evaluation.

**Learning Rate.** We conducted an ablation study on the learning rate for PEZ and two other discrete optimizers. As shown in Figure 5(a), PEZ is robust and reliable across a wide range of learning rates, from $0.001$ to $100$. In contrast, *AutoPrompt$_{SGD}$* and *FluentPrompt* are sensitive to the choice of the learning rate, with performance approaching that of a random prompt at low learning rates.

**Prompt Length.** We further ablate the optimal number of tokens. In Figure 5(b), we find that longer prompts do not necessarily produce better results when generating with Stable Diffusion, even

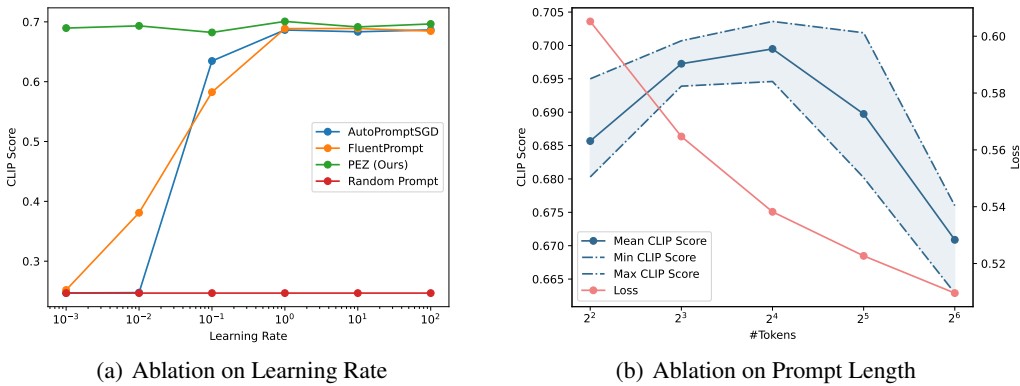

(a) Ablation on Learning Rate

(b) Ablation on Prompt Length

Figure 5: Ablation on learning rate and prompt length.

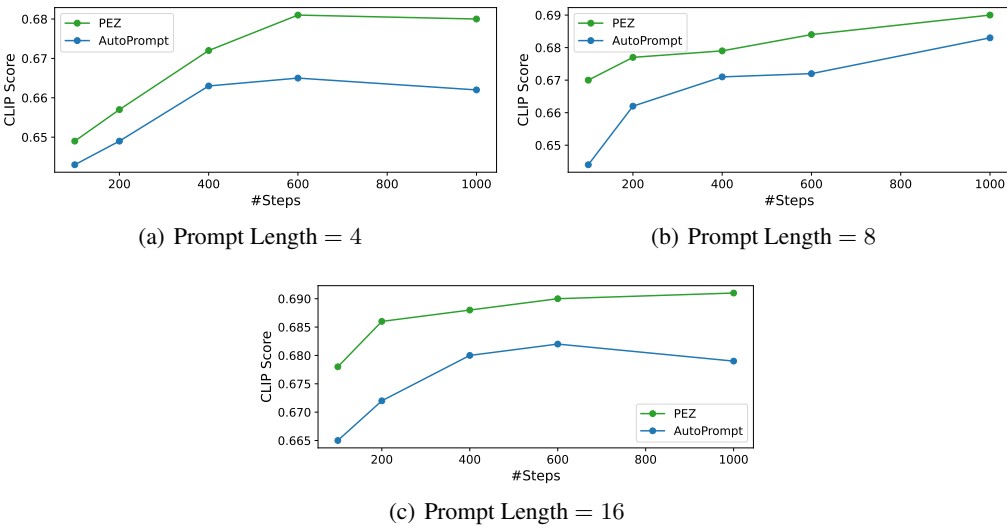

(a) Prompt Length = 4

(b) Prompt Length = 8

(c) Prompt Length = 16

Figure 6: Ablation on the number of optimization steps. For each data point, we select the maximum number from three learning rates: 0.1, 1.0, and 10.

though they strictly reduce the loss on the CLIP image encoder. Such an overfitting problem suggests that long prompts are less transferable, and we empirically find a length of 16 to result in the most generalizable performance.

**Efficiency.** We compare the convergence rates of PEZ and *AutoPrompt*$_{\text{SGD}}$ at prompt lengths of 4, 8, and 16 across three distinct learning rates: 0.1, 1, and 10. For every data point, the maximum value from the three learning rates is selected. As illustrated in Figure 6, our observations suggest that PEZ converges faster than *AutoPrompt*$_{\text{SGD}}$ for images from LIAON data. This rapid convergence is advantageous for users, allowing them to achieve superior results with fewer optimization steps.

### 4.3 Style Transfer

The proposed approach can also be easily adapted to style transfer. We follow the setting investigated with soft prompts in Gal et al. [2022] but with our hard prompts. Given several examples that share the same style, we extract their shared style characteristics into a single hard prompt and use this prompt to apply the style to new objects or scenes. Figure 3 presents two examples of style transfer, showing that our method can easily embed the shared style elements in the prompt and apply them to novel concepts. Templates and learned prompts can be found in supplementary material.

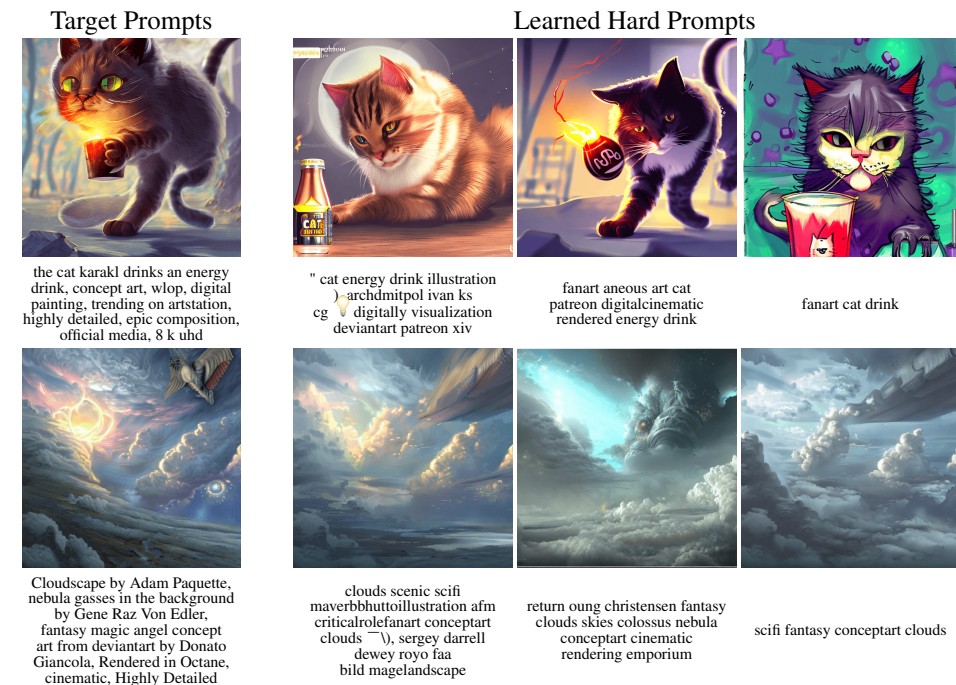

| Target Prompts | Learned Hard Prompts |
| --- | --- |

the cat karakl drinks an energy drink, concept art, wlop, digital painting, trending on artstation, highly detailed, epic composition, official media, 8 k uhd

" cat energy drink illustration ) archdmitpol ivan ks cg 👇 digitally visualization deviantart patreon xiv

fanart aneous art cat patreon digitalcinematic rendered energy drink

fanart cat drink

Cloudscape by Adam Paquette, nebula gasses in the background by Gene Raz Von Edler, fantasy magic angel concept art from deviantart by Donato Giancola, Rendered in Octane, cinematic, Highly Detailed

clouds scenic scifi maverbbhuttoillustration afm criticalrolefanart conceptart clouds ⁻\), sergey darrell dewey royo faa bild magelandscape

return oung christensen fantasy clouds skies colossus nebula conceptart cinematic rendering emporium

scifi fantasy conceptart clouds

Figure 7: **Prompt distillation**. With fewer tokens, the hard prompts can still generate images very similar in concept to the original.

## 4.4 Prompt Concatenation

Learned hard prompts are also very useful as composable building blocks for intricate scenes. We test this in Figure 4, where we separately generate prompts for two unrelated images, and then fuse both images by concatenating their prompts. We find that even different concepts, such as painted horses on a beach and a realistic sunset in a forest can be combined via their generated prompts.

## 4.5 Prompt Distillation

Another application where we can use our prompt optimization method is prompt distillation, reducing the length of prompts while preserving their capability. Distillation is useful in situations where the text encoder of the diffusion model has a limited maximum input length, such as the CLIP model, which has a maximum input length of 77 tokens. Also, long prompts may contain redundant and unimportant information, especially when hand-crafted, so we aim to distill their essence, preserving only important information in the prompt. We optimize a shorter prompt to match the features of the longer prompt simply based on its text encoder $f$. Given a target prompt's embedding $\mathbf{P}_{\text{target}}$ and learnable embedding $\mathbf{e}$, we simply modify our loss into: $\mathcal{L} = 1 - Sim(f(\mathbf{P}_{\text{target}}), f(\mathbf{P}))$. We define the distillation ratio by $|\mathbf{P}|/|\mathbf{P}_{\text{target}}|$, where $|\mathbf{P}|$ is the number of tokens in the prompt.

In Figure 7, we show images generated by the original prompts and the distilled prompts with four different distillation ratios: $0.5$, $0.3$, and $0.1$. We see here that even with only $3$ or $4$ tokens, the hard prompts can still generate images very similar in concept to the original, successfully distilling the longer human-made instructions. We further show the quantitative results of prompt distillation in supplementary material Figure 9. The distilled prompts can still maintain high CLIP scores even if the ratio is below $0.3$.

## 5 Safety Concerns

Token or word-level content filters are often used in text-to-image diffusion model APIs to prevent the generation of NSFW or copyrighted content. For instance, the image generation API `Midjourney`

has banned prompts containing the substring "Afghan" due to a copyright issue with the famous photo of an Afghan girl [2].

However, prompt optimization can be used as a mechanism to bypass simple rule-based content filters. PEZ can generate a prompt that avoids banned tokens, yet still matches textual features with the original target prompt "Afghan girl." Figure 8 shows the output of `Midjourney` using an optimized prompt which successfully reproduces the banned image without containing the banned word "Afghan." Note that the prompt seems to incorrectly associate the subject of the image, Sharbat Gula, with the Taliban.

Even if a defender now iterates the block-list and bans additional words from the adversarial prompt, an attacker can consistently optimize around addition content restrictions, as we show in supplementary material Figure 11. Overall, we suspect that only complete feature-based content detectors have the potential to mitigate these concerns for model owners [Rando et al., 2022].

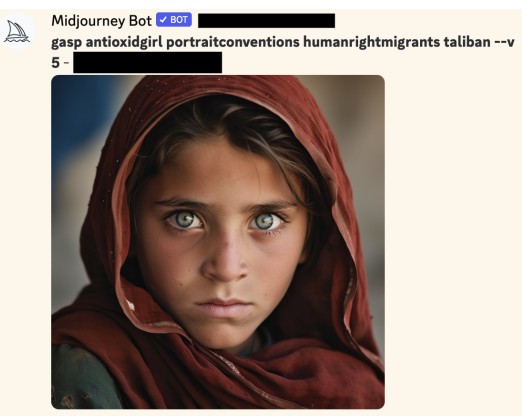

Figure 8: Generated copyrighted image via `Midjourney`. Here, requested from the API only for research purposes.

# 6 Conclusion

Overall, we show through our experiments that hard prompts can be easily generated and flexibly used in practical applications. To make hard prompts easy, we propose a new variant that utilizes continuous embeddings to reliably optimize hard prompts. The key advantage of this method, PEZ, is the use of continuous, i.e. soft, prompts as intermediate variables during the optimization of hard prompt tokens, leveraging gradient-based optimization.

Hard prompts are helpful for users of a number of image generation systems because they are easy to understand, edit, extend, and combine with existing concepts. Yet, a limitation of hard prompts is that even though they are human-readable, they may still contain several un-interpretable tokens. Additionally, hard prompts can surface harmful phrases or sensitive content from a model's training data.

# 7 Acknowledgements

This work was made possible by the ONR MURI program, the Office of Naval Research (N000142112557), and the AFOSR MURI program. Commercial support was provided by Capital One Bank, the Amazon Research Award program, and Open Philanthropy. Further support was provided by the National Science Foundation (IIS-2212182), and by the NSF TRAILS Institute (2229885).

---

[2] https://en.wikipedia.org/wiki/Afghan_Girl

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

# A    Appendix

## A.1   Additional Results for Prompt Inversion with CLIP

We provide more qualitative results in Figure 10.

For each example in Figure 3, we use the following templates respectively: "a tiger in the style of {}", "the streets of Paris in the style of {}", "a rocket in the style of {}", where {} is replaced with the hard prompts:

```
resonvillains stargazing illustration tutorials sma internationalwomensday
watercolor fiberlilycamila yokohama -sorrow fluids latest
```

```
npr anime novels pureibanganesha irvin paints encapsulmondo
illustrillustroversized sultanconan ¢
```

for experiments 1 and 2, respectively.

Table 2: Quantitative results on learned hard prompts. We report the CLIP score between the original images and the images generated by the hard prompts.

| Method | #Tokens | Requirement | LAION | MS COCO | Celeb-A | Lexica.art |
|---|---|---|---|---|---|---|
| AutoPrompt$_{\text{SGD}}$ | 8 | CLIP | $0.689_{\pm 0.001}$ | $0.669_{\pm 0.003}$ | $0.595_{\pm 0.001}$ | $0.702_{\pm 0.001}$ |
| FluentPrompt | 8 | CLIP | $0.688_{\pm 0.001}$ | $0.671_{\pm 0.005}$ | $0.583_{\pm 0.004}$ | $0.702_{\pm 0.002}$ |
| PEZ (Ours) | 8 | CLIP | $0.697_{\pm 0.001}$ | $0.677_{\pm 0.001}$ | $0.602_{\pm 0.003}$ | $0.711_{\pm 0.002}$ |
| CLIP Inter. | $\sim 77$ | C. + Ba. + BL. | 0.707 | 0.690 | 0.558 | 0.762 |
| PEZ + Bank | 8 | CLIP + Bank | $0.702_{\pm 0.001}$ | $0.689_{\pm 0.001}$ | $0.629_{\pm 0.003}$ | $0.740_{\pm 0.001}$ |
| PEZ + 5 Seeds | 8 | C. + 5 Seeds | 0.705 | 0.692 | 0.614 | 0.722 |
| C. I. w/o BLIP | $\sim 77$ | CLIP + Bank | 0.677 | 0.674 | 0.572 | 0.737 |
| CLIP Inter. | 8 | C. + Ba. + BL. | 0.539 | 0.575 | 0.360 | 0.532 |
| CLIP Inter. | 16 | C. + Ba. + BL. | 0.650 | 0.650 | 0.491 | 0.671 |
| CLIP Inter. | 32 | C. + Ba. + BL. | 0.694 | 0.663 | 0.540 | 0.730 |
| Soft Prompt | 8 | CLIP | 0.408 | 0.420 | 0.451 | 0.554 |

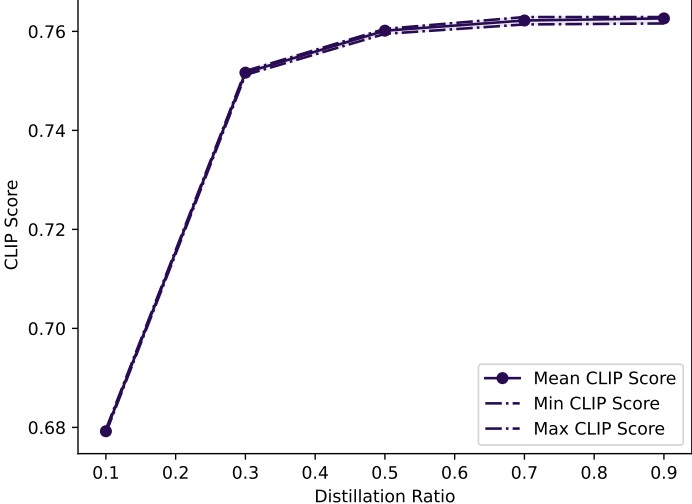

Figure 9: Quantitative results on prompt distillation with different distillation ratios. The CLIP score is calculated between the images generated by the original prompt and the images generated by the distilled prompt.

Target Image           Generated Image with Learned Hard Prompt

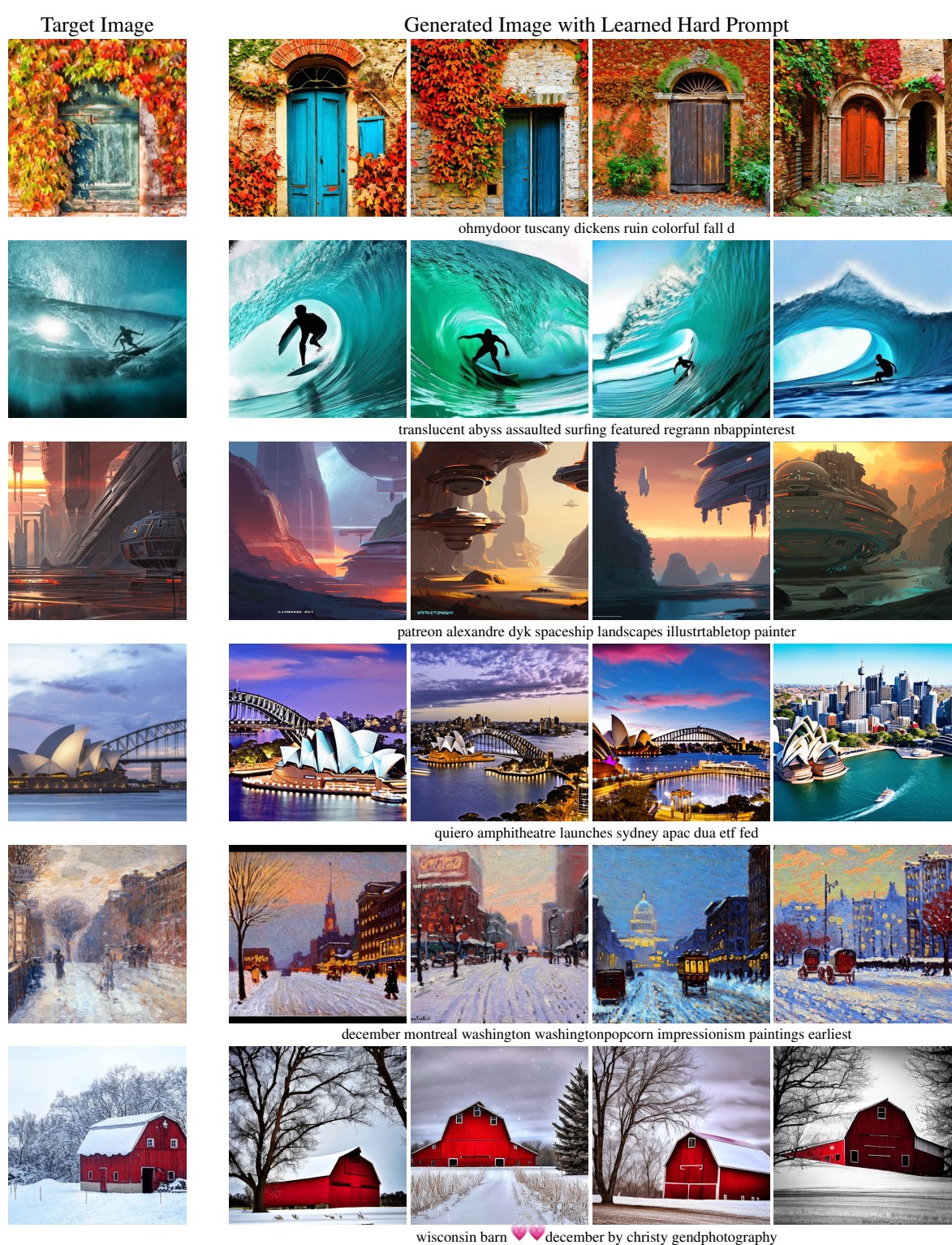

ohmydoor tuscany dickens ruin colorful fall d

translucent abyss assaulted surfing featured regrann nbappinterest

patreon alexandre dyk spaceship landscapes illustrtabletop painter

quiero amphitheatre launches sydney apac dua etf fed

december montreal washington washingtonpopcorn impressionism paintings earliest

wisconsin barn 💜💜december by christy gendphotography

Figure 10: Additional qualitative results with learned hard prompts.

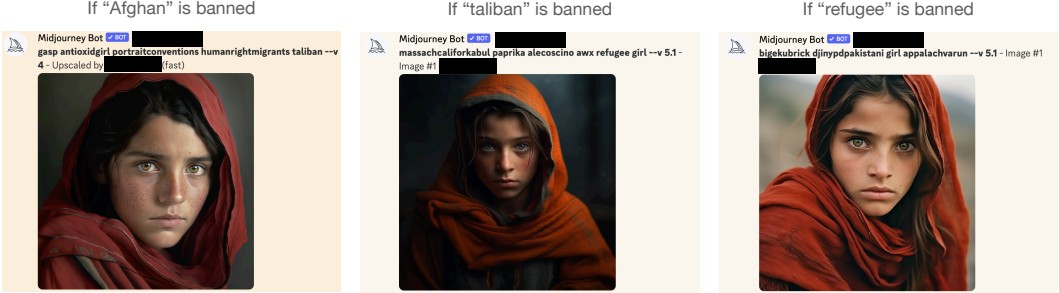

Figure 11: Iteratively evade `Midjourney` content filter and remove sensitive words/tokens.

## A.2 Text-to-Text Experiments

In this section, we compare our Algorithm 1 to its counterparts in text-to-text setting, which is the more classical setting. We found that PEZ is comparable to other gradient methods outperforming on the classification dataset, AGNEWS. Furthermore, we found that PEZ transfers better.

Table 3: Accuracy and standard deviation on the SST-2 validation set across the fives prompts for each method trained on GPT-2 Large and transferred onto larger models ranging from 1.3B to 6.7B. The baseline accuracy of a *soft prompt* is $93.35_{\pm 0.01}$ (optimized for GPT-2 Large), but cannot be transferred. Note Empty$_{\text{Template}}$ refers to no prompt at the front but containing the predetermined template.

| Method | GPT-2 Large (755M, **Source**) | GPT-2 XL (1.3B) | T5-LM-XL (3B) | OPT (2.7B) | OPT (6.7B) |
|---|---|---|---|---|---|
| Empty$_{\text{Template}}$ | 80.84 | 73.85 | 52.75 | 72.48 | 58.72 |
| AutoPrompt$_{\text{SGD}}$ | $87.56_{\pm 0.48}$ | $78.19_{\pm 6}$ | $56.01_{\pm 3.74}$ | $73.69_{\pm 3.64}$ | $65.28_{\pm 3.91}$ |
| FluentPrompt | $\mathbf{88.33}_{\pm 0.48}$ | $78.53_{\pm 6.3}$ | $55.64_{\pm 1.33}$ | $70.39_{\pm 4.66}$ | $61.74_{\pm 2.8}$ |
| Ours$_{\text{No Fluency}}$ | $88.12_{\pm 0.21}$ | $77.8_{\pm 7.71}$ | $61.12_{\pm 6.57}$ | $76.93_{\pm 2.88}$ | $71.72_{\pm 7.06}$ |
| Ours$_{\text{Fluency}}$ | $88.05_{\pm 0.76}$ | $\mathbf{79.72}_{\pm 7.3}$ | $\mathbf{63.3}_{\pm 5.14}$ | $\mathbf{77.18}_{\pm 8.54}$ | $\mathbf{72.39}_{\pm 4.07}$ |

In the context of prompting in the text-to-text setting, the goal of Algorithm 1 is to discover a discrete sequence of tokens, the hard prompt, that will prompt the language model to predict the outcome of a classification task. As an important property of text is its fluency, Shi et al. [2022] find that fluency can increase a prompt's readability and performance. Thus, we define the optimization objective in this section as a weighted function of task loss and fluency loss,

$$\mathcal{L} = (1 - \lambda_{\text{fluency}})\mathcal{L}_{\text{task}} + \lambda_{\text{fluency}}\mathcal{L}_{\text{fluency}}.$$

We set $\lambda = 0.003$ similar to Shi et al. [2022] for all methods, and we ablate our method without fluency ($\lambda = 0$), which we denote as *no fluency*. We set out to show that hard prompts generated by this approach are successful both when transferring between a number of transformer-based language models, and when used to discover prompts in few-shot settings. An attractive quality of these prompts, especially for language applications, is that they can be optimized on smaller language models and then transferred to other, much larger models.

## A.3 Datasets and Setup

We evaluate Algorithm 1 against related algorithms on three classification tasks, two sentiment analysis tasks, SST-2 [Socher et al., 2013] and Amazon Polarity [McAuley and Leskovec, 2013], and a 4-way classification task, AGNEWS [Zhang et al., 2015]. We build on the setting explored in Ding et al. [2022] and optimize hard prompts using GPT-2 Large (774M parameters) [Radford et al., 2019] with the Adafactor optimizer [Shazeer and Stern, 2018] and a batch size of 32 [Lester et al., 2021a].

**Transferability Set-up.** To test transferability, we generate prompts from GPT-2 Large for 5000 steps. We then select the five prompts with the highest average validation accuracy for each technique and test them on larger models. We test the transferred text on: GPT-2 XL, T5-LM-XL, OPT-2.7B, and OPT-6B [Radford et al., 2019, Lester et al., 2021b, Zhang et al., 2022], verifying the reliability

of the proposed algorithm over related techniques and testing whether the hard prompt can reliably boost performance. Thus, we also consider a baseline of empty prompts, with only the template.

**Few-Shot Setup.** For the few-shot setting, we optimize each prompt for 100 epochs on GPT-2 Large on the AGNEWS dataset, where we sample two examples ($k = 2$) and four examples ($k = 4$) from each class to obtain the training set. Additionally, we create a holdout set of the same size, and finally validate the prompts on the entire validation set.

## A.4 Results

We verify that our method is comparable to other methods in the sentiment analysis setting outperform the other methods on AGNEWS by about $2\%$. See Table 4 for details.

For Table 4, we report the best validation accuracy across three learning rates (0.1, 0.3, and 0.5), and for *FluentPrompt* and *AutoPrompt*$_{SGD}$ we used the learning reported (1, 3, and 10) and follow Shi et al. [2022] for the remaining hyperparameters for *FluentPrompt*. For these experiments, we *prepend* our 10 token prompt to each input text. We employ early stopping for all methods using a hold-out set of 5000 examples for each dataset, evaluating every 100 steps.

Table 4 shows that we are comparable to other methods in sentiment analysis and outperform the other methods on AGNEWS by about $2\%$. Examining the prompts, we find prompts are not coherent English for any of the methods. However, it does produce relevant tokens and phrases. For example, our method for SST-2 with the fluency constraint produced "*negative vibeThis immatureollywood MandarinollywoodThis energetic screenplay.*" [3] This suggests the optimization process is finding relevant words to the task but lacks the ability to create full sentences.

Table 4: Validation accuracy for 10 discrete tokens trained **prepended at the beginning of the input text**. Best accuracy across three learning with standard error reported over 5 speeds.

| Method | SST-2 | AGNEWS | Amazon |
|---|---|---|---|
| AutoPrompt$_{SGD}$ | $87.56_{\pm0.35}$ | $74.36_{\pm0.47}$ | $87.75_{\pm0.17}$ |
| FluentPrompt | $\mathbf{88.33}_{\pm0.35}$ | $74.62_{\pm0.24}$ | $87.42_{\pm0.18}$ |
| Ours$_{\text{No Fluency}}$ | $88.12_{\pm0.15}$ | $\mathbf{77.06}_{\pm0.20}$ | $87.70_{\pm0.21}$ |
| Ours$_{\text{Fluency}}$ | $88.05_{\pm0.55}$ | $76.94_{\pm0.48}$ | $\mathbf{87.78}_{\pm0.19}$ |
| Soft Prompt | $93.35_{\pm0.01}$ | $92.76_{\pm0.01}$ | $94.65_{\pm0.01}$ |

**Prompt Transferability.** Table 3 shows for each method the five prompts trained on GPT-2 Large transferred to other LLMs. Interestingly, simply scaling a model–with no additional training–does not guarantee that the model will scale perform according on SST-2.[4] We see that all gradient-based methods are able to transfer compared to evaluating just the template, finding that our prompts trained with the fluency constraint transfer better than the other prompts. Additionally, we can see the largest boost from OPT-6.7B with our fluent method with about a $14\%$ increase over just the template baseline. Additionally, we see our AGNEWS prompts are able to transfer from GPT-2 Large to GPT-2 XL in Table 5.

Table 5: Shows the validation accuracy with standard deviation from transferring hard prompts learned on GPT-2 Large to GPT-2 XL.

| Method | GPT-2 Large (755M) | GPT-2 XL (1.3B) |
|---|---|---|
| Empty$_{\text{template}}$ | 58.34 | 52.42 |
| AutoPrompt | $74.36_{\pm0.47}$ | $63.79_{\pm3.61}$ |
| FluentPrompt | $74.62_{\pm0.24}$ | $61.57_{\pm5.1}$ |
| Ours$_{\text{No Fluency}}$ | $77.06_{\pm0.20}$ | $59.45_{\pm8.63}$ |
| Ours$_{\text{Fluency}}$ | $76.94_{\pm0.48}$ | $67.59_{\pm2.67}$ |

---

[3]Although we initialize the tokens with the label tokens, when examining the prompt over the optimization process, all tokens moved away from the initial tokens. This suggests that the process was able to relearn the class label.

[4]A quick experiment with and without the template on GPT-2 Large and XL showed that the template boosts performance differently for different models.

Table 6: Average validation accuracy with standard error on AGNEWS with $k$ examples/shots per class using early stopping (including soft prompt) for all methods across 100 seeds for three tokens **append to the end of the text** similar to the original template ("It was about"). We set $\lambda = 0.03$ for these experiments. "Empty" is the template with no additional prompt.

| Method | $k$=2 | $k$=4 |
|---|---|---|
| Empty$_{\text{Template}}$ | 58.34 | 58.34 |
| Ours$_{\text{No Fluency}}$ | $70.07_{\pm 0.81}$ | $73.99_{\pm 0.45}$ |
| Ours$_{\text{Fluency}}$ | $70.93_{\pm 0.60}$ | $74.15_{\pm 0.48}$ |
| Soft Prompt | $74.92_{\pm 0.58}$ | $79.93_{\pm 0.36}$ |

**Prompt Discovery.** Table 6 shows that even with just a few shots we can achieve high validation accuracy compared to our prepended counterparts. It is worth noting that each few-shot run takes about 5min. We ran 100 seeds where the training set contains $k$ samples each class and did a quick examination of the top prompts, and although many of the prompts were gibberish, many of them were coherent. For example, even for $k = 2$, some of the prompts included news sources like "*BBC*", while other prompts found new approaches to the news classification task considering the text coming from a blog: "*Brian blog,*" or "*Blog Revolution analyze*." Due to the efficiency of these gradient-based methods, these methods can allow new ways for prompt engineers to discover novel prompts.

