# OpenReview forum: "Hard Prompts Made Easy: Gradient-Based Discrete Optimization for Prompt Tuning and Discovery"
_NeurIPS.cc/2023/Conference — NeurIPS 2023 poster_

### Official Review · Reviewer_7Vg6 · 2023-06-29

**Soundness:** 4 excellent
**Presentation:** 4 excellent
**Contribution:** 4 excellent
**Rating:** 7
**Confidence:** 3

**Summary:**

The paper presents an approach to optimize hard text prompts for generative models through efficient gradient-based optimization. The method automatically generates hard text-based prompts for both text-to-image and text-to-text applications, allowing users to easily generate, discover, and mix and match image concepts without prior knowledge on how to prompt the model. The paper highlights the advantages of hard prompts over soft prompts and demonstrates the effectiveness of the approach in tuning language models for classification.



**Strengths:**

1. The paper's approach is efficient and can be optimized on smaller language models and then transferred to other, much larger models.

2. The method performs consistently across all four datasets and outperforms other gradient-based optimization baselines. It can achieve similar performance to CLIP Interrogator. However, the proposed method only uses the CLIP model for prompt discovery and 8 tokens in total demonstrating its simultaneous simplicity and strength.

3. The proposed approach can also be easily adapted to style transfer. Given several examples that share the same style, the method can extract their shared style characteristics into a single hard prompt and use this prompt to apply the style to new objects or scenes (Page 5).


**Weaknesses:**

1. While the method can generate relevant tokens and phrases, the prompts are not always coherent English. This suggests that while the optimization process is finding relevant words to the task, it lacks the ability to create full sentences.

2. Longer prompts do not necessarily produce better results when generating with Stable Diffusion, even though they strictly reduce loss on the CLIP image encoder. Long prompts thus overfit and are less transferable. Thus, it may require hyperparameter tuning, which may heavy and tedious.

**Questions:**

You mentioned that longer prompts do not necessarily produce better results and can lead to overfitting. Could you elaborate on the mechanisms behind this phenomenon and suggest potential strategies to mitigate this issue while maintaining the quality of the generated prompts?

**Limitations:**

As the author has discussed, the generated prompts may still contain several un-interpretable tokens. Also, it may lack the ability to create full sentences.

---

> ### Author Rebuttal · Authors · 2023-08-09
>
> Thank you for your feedback. Below, we address specific points you raised:
>
> > You mentioned that longer prompts do not necessarily produce better results and can lead to overfitting. Could you elaborate on the mechanisms behind this phenomenon and suggest potential strategies to mitigate this issue while maintaining the quality of the generated prompts?
>
> Upon extending the prompt length, we observed an increase in the occurrence of gibberish and Unicode tokens within the optimized prompts. Consequently, this led to diminished transferability in comparison to the usage of shorter prompts. There is a transfer happening here, as we optimize on CLIP, but test on text-to-image diffusion models. If we overoptimize a prompt against CLIP, it cannot transfer to the diffusion model.
>
> We believe that introducing a constraint, such as enforcing optimized tokens to adhere to the English language, has the potential to alleviate the overfitting issue. To validate this notion, we conducted a preliminary assessment. When utilizing the keyword bank as a constraint during optimization, we noted a notable improvement in mitigating the overfitting concern, as illustrated in the table below. We will include more experiments with different constraints in the future version.
>
> |                    | 4 tokens |   8   |   16  |   32  |   64  |
> |:------------------:|:--------:|:-----:|:-----:|:-----:|:-----:|
> |         PEZ        |   0.686  | 0.697 | 0.699 | 0.689 | 0.670 |
> | PEZ + Keyword Bank |   0.684  | 0.699 | 0.712 | 0.716 | 0.708 |
>
> Thank you for your feedback on this submission. Hope our additional experiment resolves your question. Please let us know if you have other questions and comments that we can address.

---

> > ### Comment · Reviewer_7Vg6 · 2023-08-12
> > **Acknowledgement of rebuttal**
> >
> > Thank you for the rebuttal. My questions were addressed . Thus I keep my score. Good luck!

---

### Official Review · Reviewer_DXf6 · 2023-07-04

**Soundness:** 2 fair
**Presentation:** 2 fair
**Contribution:** 2 fair
**Rating:** 6
**Confidence:** 3

**Summary:**

The paper presents an easy-to-use approach to automatically obtain hard prompts for images. The work introduces PEZ, a gradient-based approach to obtain hard prompts for images. The experiments comparing a popular baseline show improved CLIP score. Finally, the qualitative results show that the method can distill prompts into shorter sequences and concatenates

**Strengths:**

The paper is easy to follow. The algorithm is well explained and the results are quite promising. The results for prompt inversion appear to be fair and outperform the baseline. The analysis is quite clear and thorough. Overall, the work presents a straightforward method to get prompt tokens for images.

**Weaknesses:**

The contribution of the work is limited: the work is mostly an application of existing work that discretizes soft prompts to tokens [a,b]. There is no significant difference between the proposed work and existing related work. The authors should clarify, in-depth, the main differences between their work and other work. Further, like AutoPrompt, the work suffers from gibberish prompts according to the authors (see Figure 2, “uuuu romantic canvas impressionist …”). This limitation makes the hard prompts harder to interpret. The explanation regarding the emoji appearing in the hard prompt (Figure 2 and line 174) is not entirely convincing. It would be really helpful to also know how often the model produces gibberish.

Limited Evaluation: the experiments do not overwhelming support that PEZ is better than CLIP interrogator. On two out of the four datasets under consideration, CLIP interrogator achieves a higher CLIPScore. It is also unclear why AutoPromptSGD is not included in Table 1. Wouldn’t this be a good baseline?

Negative Qualitative Examples: The paper could benefit from showing negative results where the method doesn’t perform as well.

Nit:
It would be helpful to look at the results along with Tables/Figures on the same page. At the moment, style transfer, prompt concatenation, and prompt distillation do not appear on the same page.

[a] Gradient-Based Constrained Sampling from Language Models. EMNLP 2022.

[b] Toward Human Readable Prompt Tuning: Kubrick’s The Shining is a good movie, and a good prompt too? ArXiv 2022.


**Questions:**

Please address the questions in the weakness section.

Minor: since you are using cosine distance as a loss function, does it run into instability issues? Is there a particular reason you are using cosine distance as a loss function?


**Limitations:**

The method requires the model weights to produce hard prompts. Given that recent methods use APIs, it would be useful to highlight this as a potential limitation.

---

> ### Author Rebuttal · Authors · 2023-08-09
>
> Thank you for your feedback. We address each of your points below:
>
> > 1. The contribution of the work is limited.
>
> The changes between **PEZ** and $FluentPrompt$ may appear subtle, but they are important for performance, and the comparison can be found in the appendix. **PEZ's** main difference is how the rounding is done and can be thought of as the combination of optimizing soft prompts and $AutoPrompt_\text{SGD}$. $AutoPrompt_\text{SGD}$ updates the set of tokens at every step using these same tokens to get the gradient information for the update. Soft prompts optimize the same continuous set of embeddings $P$ which is used for every forward and backward pass and the update step $P=P-\eta\nabla_{\mathbf{P}}\mathcal{L}$. FluentPrompt adds a decaying noise term during the optimization process.
>
> In contrast, **PEZ** is like optimizing a soft prompt, but for each gradient update the information is going to come from the nearest-neighbor rounding prompt $P'=\text{Proj}(P)$ instead of soft prompt, $P$. Then, the update is on $P$ not $P'$, $P=P-\eta\nabla_{\mathbf{P'}}\mathcal{L}$, where $P'=\text{Proj}(P)$. Thus, the process is storing a continuous set of embedding that is required for the forward pass. This means that although it may suffer from the stochastic rounding issues in the earlier steps, where the tokens project to the same set of tokens, eventually **PEZ** will find a new set of tokens since $P$ in a sense is summing the gradient information at every step. The problem with FluentPrompt and $AutoPrompt_\text{SGD}$ is sensitive to learning rates, but PEZ is quite robust under different learning rates as shown in Figure 5a. Conceptually, this tries to solve the problem that certain models require some interesting (and non-obvious) learning rates that can greatly vary from model to model.
>
> Additionally, we have now conducted a series of experiments comparing the efficiency between PEZ and AutoPrompt$_\text{SGD}$ using varying numbers of steps, as outlined in the table below. Our findings indicate that PEZ achieves faster convergence compared to AutoPrompt.
>
> |     Method     | 100 Steps |  200  |  400  |  600  |  800  |  1000 |
> |:--------------:|:---------:|:-----:|:-----:|:-----:|:-----:|:-----:|
> | AutoPrompt     |   0.642   | 0.654 | 0.656 | 0.672 | 0.677 | 0.684 |
> |       PEZ      |   0.668   | 0.674 | 0.675 | 0.684 | 0.690 | 0.695 |
>
> We believe that both properties of PEZ are very important to the prompt optimization applications mentioned in the paper.
>
> >Limited Evaluation: the experiments do not overwhelmingly support that PEZ is better than CLIP interrogator.
>
> We have updated our local draft by noting that the CLIP interrogator is a purpose-built tool, built to generate fitting prompts for digital art images, from a word bank of possible artistic styles and expressions. That the proposed generic algorithm is able to match the performance of this tool is a strong statement of its general capability.  We will include these updates in our camera ready version.
>
> The issues with using the CLIP Interrogator in any other context than generating captions for art is highlighted in the difference we see in the Celeb-A dataset, where PEZ outperforms CLIP Interrogator as these images is "OOD" for both the BLIP model and the small word bank. Additionally, due to the compressibility of using PEZ, we can combine different prompts together as highlighted in Sections 4.3 and 4.4. However, under the current functionality of the CLIP Interrogator, this is not a possibility. In short, PEZ outperforms in recreating images that fall outside the CLIP Interrogator's prior and has additional functionality, while matching this purpose-built tool on its exact task.
>
> A full table of these comparisons can be found in Table 2 in the Supplementary Material.
>
>
> > Negative Qualitative Examples.
>
> We also think it's helpful to show some failure examples. We have added some failure generations in the rebuttal PDF. As we explained in the experiment part, the results on the Celeb-A dataset aren't as good as on other datasets. This might be because learning to make human portraits is tough, especially for small details in the face. As we can see in Figure 1 in the rebuttal PDF, the prompts from PEZ can still make pictures of human faces that look similar to the target images, but not exactly the same.
>
> Meanwhile, we provide other two typical failure cases in Figure 1: 1) instances where the generated images exhibit distinct styles from the target image (the second-to-last example); 2) cases where the prompt captures abstract concepts rather than specific objects (the final example). It's important to note, however, that these two scenarios can be mitigated by optimizing with various initializations.
>
> We will include more failure examples and a comprehensive discussion and investigation of these cases in our future version.
>
> > Since you are using cosine distance as a loss function, does it run into instability issues?
>
> CLIP used cosine similarity to train the model, so we actually did not suffer from these instability issues as we used the same objective function.
>
> > The method requires the model weights to produce hard prompts. Given that recent methods use APIs, it would be useful to highlight this as a potential limitation.
>
> This is a great point and we have extended our discussion of this limitation. However, many API models are based on publically-available CLIP checkpoints, for example, Midjourney. A great example of this process can be found in Section 5, where we show that in fact, we can bypass the content filters of the Midjourney API using optimized prompts created with open-source CLIP weights.
>
> Thank you again for your thoughtful review. We made a significant effort to address your feedback including multiple paper edits, and we would appreciate it if you would consider raising your score in light of our response. Do you have any additional questions we can address?

---

> > ### Comment · Reviewer_DXf6 · 2023-08-16
> > **Reply to the Authors**
> >
> > Thank you for the detailed response. I really appreciate you uploading additional negative results in the PDF. I also liked your response to Reviewer LSBN where you have included prompts with more tokens (8 to 64). It would be awesome if you could include this ablation in the paper as well.
> >
> > I do have a few concerns:
> >
> > 1. The additional table included in the rebuttal shows that PEZ converges faster than AutoPrompt. This is a positive result. I would include this detail in the paper. But, as pointed out in my review, Table 1 in the paper is incomplete without AutoPrompt. It might be the case that AutoPrompt performs better than PEZ on all the other datasets. If the work is about efficiency compared to AutoPrompt, then you might have to highlight the tradeoffs between PEZ and AutoPrompt in terms of efficiency vs. performance.
> >
> > 2. Minor: could you clarify what you mean by the following statement?
> > > It's important to note, however, that these two scenarios can be mitigated by optimizing with various initializations.
> >
> > Overall, I think the authors have put in a lot of effort to improve their paper. However, I feel the paper could be positioned better to highlight the practical benefits in terms of efficiency compared to existing work. I think the work would benefit from another round of submission. I will gladly increase my scores if other reviewers feel strongly about the paper.

---

> > > ### Author Response · Authors · 2023-08-18
> > > **Official Comment by Authors**
> > >
> > > Regarding 1, we included the results for AutoPrompt in Table 2 of the Appendix. PEZ shows improvement over AutoPrompt. We have updated our working draft to incorporate these findings, as well as the efficiency results like the table below, into the main paper and will include them in the camera-ready version.
> > >
> > > | prompt length |  lr |   method   | 100 steps |  200  steps |  400 steps |  600 steps |  1000 steps |
> > > |:-------------:|:---:|:----------:|:---------:|:-----:|:-----:|:-----:|:-----:|
> > > |       4       | 0.1 | AutoPrompt |   0.461   | 0.515 | 0.536 | 0.551 | 0.578 |
> > > |               |     |     PEZ    |   0.610   | 0.635 | 0.653 | 0.666 | 0.666 |
> > > |               |  1  | AutoPrompt |   0.614   | 0.639 | 0.663 | 0.667 | 0.670 |
> > > |               |     |     PEZ    |   0.649   | 0.657 | 0.663 | 0.660 | 0.668 |
> > > |               |  10 | AutoPrompt |   0.643   | 0.649 | 0.663 | 0.665 | 0.662 |
> > > |               |     |     PEZ    |   0.645   | 0.653 | 0.672 | 0.681 | 0.680 |
> > > |       8       | 0.1 | AutoPrompt |   0.516   | 0.553 | 0.582 | 0.598 | 0.614 |
> > > |               |     |     PEZ    |   0.632   | 0.656 | 0.671 | 0.677 | 0.678 |
> > > |               |  1  | AutoPrompt |   0.641   | 0.654 | 0.656 | 0.672 | 0.683 |
> > > |               |     |     PEZ    |   0.667   | 0.673 | 0.674 | 0.684 | 0.690 |
> > > |               |  10 | AutoPrompt |   0.644   | 0.662 | 0.671 | 0.671 | 0.676 |
> > > |               |     |     PEZ    |   0.670   | 0.677 | 0.679 | 0.679 | 0.682 |
> > > |       16      | 0.1 | AutoPrompt |   0.551   | 0.595 | 0.628 | 0.631 | 0.639 |
> > > |               |     |     PEZ    |   0.646   | 0.665 | 0.679 | 0.686 | 0.695 |
> > > |               |  1  | AutoPrompt |   0.663   | 0.672 | 0.680 | 0.680 | 0.687 |
> > > |               |     |     PEZ    |   0.657   | 0.686 | 0.678 | 0.690 | 0.691 |
> > > |               |  10 | AutoPrompt |   0.665   | 0.665 | 0.672 | 0.682 | 0.679 |
> > > |               |     |     PEZ    |   0.678   | 0.682 | 0.681 | 0.681 | 0.686 |
> > >
> > > Regarding 2, for certain failure cases, we observed that users can achieve a more effective prompt by restarting the optimization with a different initialization for the hard prompt. This situation is similar to the "+5 seeds" scenario depicted in Table 1. We see that this phrase was ambiguous, so we have clarified it in our draft, and we thank you for pointing that out.
> > >
> > > Thank you again for your detailed and constructive feedback.  We have incorporated each of your suggestions into our draft, and we would appreciate it if you would consider increasing your score accordingly.  Do you have any other questions we can address?

---

> > > > ### Comment · Reviewer_DXf6 · 2023-08-18
> > > > **Updating score to weak accept**
> > > >
> > > > Thank you for adding the additional result. First, this Table highlights three points: (1) PEZ requires fewer prompt tokens, (2) PEZ converges faster than AutoPrompt, and (3) PEZ is robust to the change in learning rate. It would be great if you could include these points in the paper. It could help better position your work in terms of efficiency. Second, you could suggest an explanation for the high learning rate (lr=10). Finally, Your most recent response to the reviewer Lu7o is helpful to the readers. You could include that detail in the paper.
> > > >
> > > > I am updating my score from 4 to 6.

---

### Official Review · Reviewer_LSBN · 2023-07-06

**Soundness:** 3 good
**Presentation:** 2 fair
**Contribution:** 3 good
**Rating:** 5
**Confidence:** 3

**Summary:**

This paper works on hard prompt optimization with gradient methods, especially a discrete text prompt is discovered using CLIP, and optimized to prompt stable diffusion.

**Strengths:**

1. Without hand-crafted design of hard prompt, the proposed solution directly discover and optimize prompt with gradient descent, leading to very efficient prompt engineering.
2. The learned prompt works great on stable diffusion and achieve effective style transfer, further explains the superiority of the proposed solution.

**Weaknesses:**

The proposed solution works on discrete text prompt optimization without constraints on meaningfulness of the text, making it hard to directly understand to the effectiveness of prompt optimization.

**Questions:**

1. Ablation study shows performance saturation with middle number of prompt length. More analysis is needed to explain this result.
2. Prompt distillation seems interesting to explain the prompt length issue, however, how the distillation ratio is correlated with the quality of the generated samples should also be explained further.

**Limitations:**

Yes

---

> ### Author Rebuttal · Authors · 2023-08-09
>
> Thank you for your feedback. We address each of your points below:
>
> #### Weaknesses
>
> > The proposed solution works on discrete text prompt optimization without constraints on meaningfulness of the text, making it hard to directly understand to the effectiveness of prompt optimization.
>
> While we do experiment with fluency constraints, showing that it is possible to add such constraints (see experiments in the Supplementary Material), this is not actually our primary goal. We are not concerned with the fluency of the generated prompts. In all applications, a user would be able to provide any series of tokens to a generation API, and there is no need for the optimized prompt to be fluent. We agree that this is a departure from some of the literature on prompt optimization, which is concerned with finding interpretable prompts, but for the use cases we consider, interpretability is not required.
>
> For the CLIP experiments, our study also provides interesting prompts that hint at a secret language within the diffusion models. We further highlight the "secret language" in Section 5 (Safety Concerns), where are able to bypass the Midjourney content filters. In safety questions, the ability to bypass a word filter with a non-interpretable prompt is a core necessity of the attack.
>
> #### Questions:
>
> Regarding Q1, our suspicion is rooted in overfitting, evident from the decreasing loss accompanied by a lower CLIP Score. Additionally, upon extending the prompt length, we observed an increase in the occurrence of model-specific unicode token usage within the optimized prompts. Consequently, this led to diminished transferability in comparison to the usage of shorter prompts.
>
> We believe that introducing a constraint, such as enforcing optimized tokens to adhere to the English language, has the potential to alleviate the overfitting issue. To validate this notion, we have now conducted a preliminary assessment. When utilizing the keyword bank as a constraint during optimization, we noted a notable improvement in mitigating the overfitting concern, as illustrated in the table below.
>
> |                    | 4 tokens |   8   |   16  |   32  |   64  |
> |:------------------:|:--------:|:-----:|:-----:|:-----:|:-----:|
> |         PEZ        |   0.686  | 0.697 | 0.699 | 0.689 | 0.670 |
> | PEZ + Keyword Bank |   0.684  | 0.699 | 0.712 | 0.716 | 0.708 |
>
> For Q2, we explore this question in Figure 8 in Supplementary Material. We find here that this is a step down in performance from a ratio of 0.3 to 0.1.
>
> Thank you again for your thoughtful review. We made a significant effort to address your feedback including experiments, and we would appreciate it if you would consider raising your score in light of our response. Do you have any additional questions we can address?

---

### Official Review · Reviewer_Lu7o · 2023-07-07

**Soundness:** 3 good
**Presentation:** 3 good
**Contribution:** 2 fair
**Rating:** 5
**Confidence:** 4

**Summary:**

This work proposes a new paradigm which optimizes discrete prompts simply by gradient projection to bridge the gap between relatively “easy-to-optimize” soft prompts and their hard counterparts. Their methodology is straightforward and effective in downstream text-to-image task requirements. And more specifically, comparing to existing gradient-based approaches, such as AutoPrompt, their approach is less hyper-parameter sensitive equipping with their gradient projection. All these advantages offer a new lens on how to optimize discrete prompts for text-to-image generation models, and LLMs likewise.

**Strengths:**

1. This framework is easy to implement, by simply gradient projection in the soft embedding space, and the performance seems promising compared to popular caption-based "CLIP interrogator". It does contribute another approach for such optimization problem.
2. The experiments do show the advantages of such framework, such as "the insensitiveness to hyper-parameters".
3. The paper organization is good, and the writing is well-polished.

**Weaknesses:**

1. I think adding more baselines in your main table would definitely be a plus! Currently, you mainly compare this to a heuristic-based yet popular method called "CLIP interrogator". Can you compare your approach with more diversified baselines, e.g., "AutoPrompt with different LRs", "RLPrompt", etc.. These can let readers gain a basic intuitive understandings of your method's performance against others.
2. In your main table, the improvements of yours compared to CLIP interrogator seems to be marginal, where your performance is almost the same level of performance with CLIP interrogator, even though you only include fewer tokens.
3. Besides, can you show some quantitative and qualitative examples on directly showing on the same set of images, what are the optimized prompts by CLIP interrogator, and what are the optimized prompts by yours (and perhaps, their word overlapping by distillation), such that we can enhance our understandings that your approach identify a "secret" language, like initial work [1] (in addition to possible application contributions, the secret language could also contribute to scientific research if your pipeline is good at identifying such phenomenon). Otherwise, to one of your contributions that your identified prompt is shorter, we may speculate that the prompts in CLIP interrogator can also be distilled into the same short length, and even with the same key tokens as your pipeline. Therefore, the secret language observation would be weaken. Besides, the CLIP interrogator even does not require any training with similar level of performance and even similar level of short lengths through distillation, why not we use their easier pipeline? --- Currently, figure 6 further enhances my confidence on such point as well, as your prompts include a lot of natural tokens in accordance with our common-sense understandings, such as cat images should use "cat" tokens.
4. Additionally, practitioners would be interested in your optimization efficiency as well for such algorithm to identify well-crafted prompts, comparing to many other soft/hard prompt optimization work. In terms of this, if you have better efficiency, that is also another good point, where I am also interested.

[1] Discovering the hidden vocabulary of DALLE-2, arxiv 2022

So right now, I am more viewing this work's contribution as a different perspective/approach to prompt optimization for text-to-image generation models, and even more models. Of course, that is interesting, and useful/insightful to the community. However, a key limitation would be such utilities are not quite clear alongside all current prompt optimization approaches. And the performance improvement seems to somewhat limited compared to CLIP interrogator.

Hopefully, my suggestions would be useful for you to improve your content.

**Questions:**

See weaknesses.

**Limitations:**

See weaknesses.

---

> ### Author Rebuttal · Authors · 2023-08-09
>
> Thank you for your feedback. We address each of your points below:
>
> >1. I think adding more baselines in your main table would definitely be a plus! Currently, you mainly compare this to a heuristic-based yet popular method called "CLIP interrogator". Can you compare your approach with more diversified baselines, e.g., "AutoPrompt with different LRs", "RLPrompt", etc.
>
> We include AutoPrompt (SGD) with different learning rates in Figure 5(a). Additionally, we believe RLPrompt is infeasible for optimizing images as each experiment (a single image in our case) takes between 1.5 and 4 hours according to [1]. This makes RLPrompt infeasible for this particular set of experiments. Also, please see Table 2 in the Supplementary Material for more baselines.
>
> [1] Deng, Mingkai, et al. "Rlprompt: Optimizing discrete text prompts with reinforcement learning." arXiv preprint arXiv:2205.12548 (2022).
>
> >2. In your main table, the improvements of yours compared to CLIP interrogator seems to be marginal, where your performance is almost the same level of performance with CLIP interrogator, even though you only include fewer tokens.
>
> CLIP Interrogator is a purpose-built tool that has a much stronger prior than PEZ - that we are even able to match it with a general optimization scheme, is to us an indication that our much more general-purpose optimizer works quite well.
>
> Specifically, CLIP-Interrogator uses the BLIP image-to-text model and a word bank of predefined artistic phrases to search over. Problems with using this tool in any other context than generating captions for art are highlighted in the difference we see in the Celeb-A dataset, where PEZ outperforms CLIP Interrogator as these images are "OOD" for both BLIP and the small word bank. Additionally, due to the compressibility of using PEZ, we can combine different prompts together as highlighted in Sections 4.3 and 4.4. However, under the current functionality of the CLIP Interrogator, this is impossible. In short, PEZ outperforms in recreating images that fall outside the CLIP Interragator's prior and has additional functionality, while matching this purpose-built tool on its exact task.
>
> >3. Besides, can you show some quantitative and qualitative examples of directly showing on the same set of images, what the optimized prompts by the CLIP interrogator, and what the optimized prompts by yours (and perhaps, their word overlapping by distillation), such that we can enhance our understandings that your approach identifies a "secret" language, like initial work [1] (in addition to possible application contributions, the secret language could also contribute to scientific research if your pipeline is good at identifying such phenomenon). Otherwise, to one of your contributions that your identified prompt is shorter, we may speculate that the prompts in the CLIP interrogator can also be distilled into the same short length, and even with the same key tokens as your pipeline. Therefore, the secret language observation would be weakened. Besides, the CLIP interrogator even does not require any training with a similar level of performance and even a similar level of short lengths through distillation, why not use their easier pipeline? --- Currently, figure 6 further enhances my confidence on such a point as well, as your prompts include a lot of natural tokens in accordance with our common-sense understandings, such as cat images should use "cat" tokens.
>
> Do you mind clarifying the latter part of this statement? We are a little confused by what exactly you are suggesting. Thank you!
>
> Regarding the other questions, note the differences between CLIP interrogator that we mention above. We do evaluate CLIP Interrogator for shorter prompt lengths, see Table 1.
>
> > 4. Additionally, practitioners would be interested in your optimization efficiency as well for such an algorithm to identify well-crafted prompts, compared to many other soft/hard prompt optimization work. In terms of this, if you have better efficiency, that is also another good point, where I am also interested.
>
> Thank you for the suggestion. Prompted by your feedback, we have now conducted a series of experiments comparing the efficiency between PEZ and AutoPrompt$_\text{SGD}$ using varying numbers of steps, as outlined in the table below. Our findings indicate that PEZ achieves faster convergence compared to AutoPrompt.
>
> |     Method     | 100 Steps |  200  |  400  |  600  |  800  |  1000 |
> |:--------------:|:---------:|:-----:|:-----:|:-----:|:-----:|:-----:|
> | AutoPrompt     |   0.642   | 0.654 | 0.656 | 0.672 | 0.677 | 0.684 |
> |       PEZ      |   0.668   | 0.674 | 0.675 | 0.684 | 0.690 | 0.695 |
>
> Thank you again for your thoughtful review. We made a significant effort to address your feedback including experiments. We have updated our draft accordingly and will include these updates in our camera-ready version, and we would appreciate it if you would consider raising your score in light of our response. Do you have any additional questions we can address?

---

> > ### Comment · Reviewer_Lu7o · 2023-08-12
> > **Reviewer Lu7o Replies**
> >
> > <1. baselines>
> > - From your rebuttal, I have understood what is your key advantage you want to show now. Precisely, that is the **efficiency** compared to other baselines (w/ your similar level of performance as many baselines), such as AutoPrompt, RLPrompt, etc. It would be much better for you to consider adding efficiency comparisons visualizations/discussions. Additionally, I am not sure whether RLPrompt would fail in this case, as image-level/instance-level optimization takes fewer time, which deserves some experiments as well. If it is weaker/much slower than your approach in a few cases, it would be good to include this and highlight several times in your paper, to make it more clear about your key advantages. Moreover, from your new AutoPrompt experiments, it would be good for you to add one **rigorous** figure on the efficiency comparison results, e.g., using the best setup and also the less good setup (such as other LR, prompt length in AutoPrompt), to position your approach. This would definitely add some values to your paper.
> >
> > <2. CLIP interrogator>
> > - Thanks for your clarification. It does seem that interrogator performs much weaker with fewer tokens. But as you said, it would be good for you to incorporate some efficiency studies as well. That can address my concerns on your approach vs. CLIP interrogator.
> >
> > <3. Secret Language>
> > - This part is just for one of your claims. You said you identify a secret language of text-to-image diffusion models [1]. But in the initial paper, the lexical items that they used are purely gibberish tokens, w/o any significant real-world meanings, such as "Apoploe vesrreaitais" for birds. But in your results, and also your new pdf results, you always include some natural tokens, for instance, "butterfly emoji + users', 'cruise + green render lights', "balloon relationships", “cat” for cat image, and many more shown in your paper figures. I am curious about more fine-grained studies on your generated prompts. For instance, others might suspect that when deleting your natural meaningful tokens, your generated results would fail completely, which may indicate a weaker claim of "secret language". Or in other words, perhaps, most of your used nonsense tokens are mainly search artifacts. Therefore, my comments are to say it would be better for you to provide some rigorous studies.
> >
> > I believe this point is also mentioned by reviewer DXf6, for which I agree with him/her.
> >
> > [1] Discovering the Hidden Vocabulary of DALLE-2
> >
> > <4. efficiency results>
> > - See 1, it would be better for you to provide rigorous studies to position your approach.
> >
> > For current rating, I am saying that this work should be further improved to be an interesting paper published in this conference. So I select the borderline rating, in which I would like to wait for other reviewers or ACs for final judgements.

---

> > > ### Author Response · Authors · 2023-08-18
> > > **Official Comment by Authors**
> > >
> > > Thank you for your positive and thoughtful response.
> > >
> > > > <1. baselines> and <4. efficiency results>
> > >
> > > We appreciate this suggestion.  We have updated our draft to include a rigorous comparison of clip score v.s. wall-time. Due to the rebuttal rules, we can only present a markdown table here, but we will include our associated figure in the camera ready version also including RLPrompt, which is omitted here due to its very high runtime.
> > >
> > > | prompt length |  lr |   method   | 100 steps |  200  steps |  400 steps |  600 steps |  1000 steps |
> > > |:-------------:|:---:|:----------:|:---------:|:-----:|:-----:|:-----:|:-----:|
> > > |       4       | 0.1 | AutoPrompt |   0.461   | 0.515 | 0.536 | 0.551 | 0.578 |
> > > |               |     |     PEZ    |   0.610   | 0.635 | 0.653 | 0.666 | 0.666 |
> > > |               |  1  | AutoPrompt |   0.614   | 0.639 | 0.663 | 0.667 | 0.670 |
> > > |               |     |     PEZ    |   0.649   | 0.657 | 0.663 | 0.660 | 0.668 |
> > > |               |  10 | AutoPrompt |   0.643   | 0.649 | 0.663 | 0.665 | 0.662 |
> > > |               |     |     PEZ    |   0.645   | 0.653 | 0.672 | 0.681 | 0.680 |
> > > |       8       | 0.1 | AutoPrompt |   0.516   | 0.553 | 0.582 | 0.598 | 0.614 |
> > > |               |     |     PEZ    |   0.632   | 0.656 | 0.671 | 0.677 | 0.678 |
> > > |               |  1  | AutoPrompt |   0.641   | 0.654 | 0.656 | 0.672 | 0.683 |
> > > |               |     |     PEZ    |   0.667   | 0.673 | 0.674 | 0.684 | 0.690 |
> > > |               |  10 | AutoPrompt |   0.644   | 0.662 | 0.671 | 0.671 | 0.676 |
> > > |               |     |     PEZ    |   0.670   | 0.677 | 0.679 | 0.679 | 0.682 |
> > > |       16      | 0.1 | AutoPrompt |   0.551   | 0.595 | 0.628 | 0.631 | 0.639 |
> > > |               |     |     PEZ    |   0.646   | 0.665 | 0.679 | 0.686 | 0.695 |
> > > |               |  1  | AutoPrompt |   0.663   | 0.672 | 0.680 | 0.680 | 0.687 |
> > > |               |     |     PEZ    |   0.657   | 0.686 | 0.678 | 0.690 | 0.691 |
> > > |               |  10 | AutoPrompt |   0.665   | 0.665 | 0.672 | 0.682 | 0.679 |
> > > |               |     |     PEZ    |   0.678   | 0.682 | 0.681 | 0.681 | 0.686 |
> > >
> > > > <2. CLIP interrogator>
> > >
> > > The CLIP interrogator consistently requires the same amount of time regardless of the length of the prompt. This is because it calculates the clip scores between the target image and all tokens in the prior bank, then selects the top-k to fill approximately 70 tokens. However, it's crucial to highlight that our method offers competitive prompts with significantly fewer tokens than the CLIP interrogator. This provides users with increased flexibility, allowing them to concatenate tokens or use the prompt as a style guide. Such flexibility stands as a notable advantage of our pipeline.
> > >
> > > Furthermore, quantifying the efficiency of the CLIP interrogator in terms of the number of steps can be challenging, so we have added an efficiency comparison based on wall time to our working draft and will include it in our camera-ready version.
> > >
> > > > <3. Secret Language>
> > >
> > > Prompted by your feedback, we have now conducted an experiment to see if all gibberish tokens are indeed search artifacts (not meaningful for generation). We present an example here and have added others to our working draft, including both instances where gibberish tokens are valuable and ones where they are not.
> > >
> > > Consider the following text, "translucent abyss assaulted surfing featured regrann nbappinterest", which produces a surfer in a wave tunnel found in Figure 9 of the Appendix. We find that some tokens like "nbappinterest" and "assaulted" are not necessary for good generations. However, "regrann" is crucial, despite sounding unrelated to the image at hand and despite "regrann" not being a real word. Nonetheless, "regrann" is critical for producing sensible images in numerous ways (i.e. not producing a second person, making sure the surf board is in one piece, etc.). These two words/tokens contribute to the image in ways that are non-obvious at first glance. We believe that this example and our other examples suggest that the language model extracts meaning from tokens which a human would not. We do agree that adding such examples is valuable, and we have now additionally clarified in our working draft that we are referring to the extraction of meaning from tokens which a human would not extract meaning from or where a human may extract vastly different meaning.
> > >
> > > Thank you again for engaging and for your helpful suggestions.  We would greatly appreciate it if you would consider improving your score in light of our detailed response.  Do you have any other questions we can address?

---

> > > > ### Comment · Reviewer_Lu7o · 2023-08-21
> > > > **Replies from Reviewer Lu7o**
> > > >
> > > > Thanks for your replies. I have the following comments:
> > > >
> > > > 1. Yes, it would be nice to incorporate more comparisons w.r.t. your efficiency comparisons. From now on, I am also curious about when AutoPrompt could be competitive to your approach, by tuning the hyper-parameters, e.g., more training steps. And for RLPrompt, it sounds like you may need to pay attention to its input format, e.g., how many examples are batched as the final reward calculations. Perhaps, you can also do some experiments, just for more clear and thorough baselines.
> > > >
> > > > 2. Yes, it would be nice.
> > > >
> > > > 3. For secret language section, yes, please make it much more clear in your work, to avoid any confusions or ambiguities. And more visualized examples on these phenomena are high encouraged.
> > > >
> > > > Hope my comments are informative enough for you to improve your work.
> > > >
> > > > I could raise my score as well for possible well-polished paper (I am looking forward to read the updated version), and let the ACs judge the final recommendations from our discussions.

---

### Author Rebuttal · Authors · 2023-08-09

We thank all the reviewers for their time and thoughtful feedback. We have attached a PDF containing additional figures, showing explicit failure cases of the approach as requested by one reviewer.

We would like to emphasize the significance of this work. Our proposed, general-purpose, prompt optimization method is able to achieve comparable results to the purpose-built art prompt tool, CLIP interrogator, without any prior domain-specific knowledge, image-to-caption models, or keyword banks. Additionally, we find that PEZ demonstrates that it is possible to break content filters in deployed generative AI models, such as Midjourney - a new vulnerability that we uncover immediately, without even tuning the optimization scheme.

Meanwhile, the differences between PEZ and other baselines like AutoPrompt may appear subtle, but they are crucial for performance. As shown in Figure 5 (a), PEZ is much more robust under different learning rates. Also, our additional experiments, presented in the table below, suggest that PEZ converges much faster than the baseline under the same learning conditions. We believe that the flexibility and efficiency of PEZ are very important for applications like prompt optimization, really making "hard prompts easy" to use. This has been reflected in the usage and implementation of this method in a number of applications, as well as in GUI tools for hobby users of diffusion models.

|     Method     | 100 Steps |  200  |  400  |  600  |  800  |  1000 |
|:--------------:|:---------:|:-----:|:-----:|:-----:|:-----:|:-----:|
| AutoPrompt     |   0.642   | 0.654 | 0.656 | 0.672 | 0.677 | 0.684 |
|       PEZ      |   0.668   | 0.674 | 0.675 | 0.684 | 0.690 | 0.695 |

---

### Decision · Program_Chairs · 2023-09-21

**Decision:**

Accept (poster)

**Comment:**

The paper presents an approach to optimize hard text prompts for diffusion models through efficient gradient-based optimization. The authors have done a nice job during rebuttal. After rebuttal, it received scores of 5567. The scores look nice, though 3 of the 4 reviewers only gave a confidence level of 3.

On the positive side, the proposed method is easy to implement, by simply gradient projection in the soft embedding space, and the performance seems promising compared to popular caption-based "CLIP interrogator". Also, the paper is well written, and the analysis is clear and thorough. On the other hand, during the rebuttal, the authors have provided additional results, and reviewers commented that some points needed to be further adjusted to be a nice work, e.g., the advantage in terms of efficiency, compared to other methods. Otherwise, the paper shows limited technical contribution, such as novelty and performance compared to other simpler baselines. Furthermore, the discussion on "hidden vocabulary"/"secret language" can be further polished.

Overall, the AC thinks that the merits of this paper outweigh its flaws, and would like to recommend acceptance by the end. The authors are highly encouraged to add the new results into the main paper, and revise the paper accordingly based on reviewers' detailed comments.